# Microtubule-Interfering Drugs: Current and Future Roles in Epithelial Ovarian Cancer Treatment

**DOI:** 10.3390/cancers13246239

**Published:** 2021-12-12

**Authors:** Joan Tymon-Rosario, Naomi N. Adjei, Dana M. Roque, Alessandro D. Santin

**Affiliations:** 1Department of Obstetrics, Gynecology, and Reproductive Sciences, Yale University School of Medicine, New Haven, CT 06520, USA; joan.tymon-rosario@yale.edu (J.T.-R.); naomi.adjei@yale.edu (N.N.A.); 2Marlene and Stewart Greenebaum Comprehensive Cancer Center, University of Maryland School of Medicine, Baltimore, MD 21201, USA; droque@som.umaryland.edu

**Keywords:** epithelial ovarian cancer, chemotherapy, microtubule-interfering drugs, paclitaxel, ixabepilone, microtentacles

## Abstract

**Simple Summary:**

Microtubule-interfering drugs have been used alone or in combination in the treatment of epithelial ovarian cancer. Over the years and with increasing chemoresistance to taxanes, epothilones (i.e., ixabepilone) have become of interest as alternatives to taxanes. In this review, we discuss the role of microtubule-interfering chemotherapeutic agents in treatment of newly diagnosed and recurrent ovarian cancer, as well as common mechanisms of chemoresistance. We also discuss future directions for the use of microtubule-interfering agents in ovarian cancer.

**Abstract:**

Taxanes and epothilones are chemotherapeutic agents that ultimately lead to cell death through inhibition of normal microtubular function. This review summarizes the literature demonstrating their current use and potential promise as therapeutic agents in the treatment of epithelial ovarian cancer (EOC), as well as putative mechanisms of resistance. Historically, taxanes have become the standard of care in the front-line and recurrent treatment of epithelial ovarian cancer. In the past few years, epothilones (i.e., ixabepilone) have become of interest as they may retain activity in taxane-treated patients since they harbor several features that may overcome mechanisms of taxane resistance. Clinical data now support the use of ixabepilone in the treatment of platinum-resistant or refractory ovarian cancer. Clinical data strongly support the use of microtubule-interfering drugs alone or in combination in the treatment of epithelial ovarian cancer. Ongoing clinical trials will shed further light into the potential of making these drugs part of current standard practice.

## 1. Introduction

Ovarian cancer is the most common cause of cancer death in women with gynecologic malignancy and the fifth leading cause of cancer death in the United States. In 2020, there were approximately 22,000 new cases and 14,000 cancer-related deaths in the United States [1]. Unfortunately, approximately 75% of women present with advanced stage disease at diagnosis and the majority will relapse after initial treatment, at which point mortality is high due to the acquisition of progressive chemoresistance with each recurrence [2,3]. In fact, the 5-year overall survival (OS) for women diagnosed with ovarian cancer is a dismal <45% [1]. Approximately 95% of ovarian malignancies are derived from epithelial cells with subtypes that include high-grade serous, low-grade serous, endometrioid, clear cell, and mucinous. The remaining histologic subtypes arise from other ovarian cell types (i.e., germ cell or sex cord-stromal tumors) [4]. High-grade serous epithelial ovarian carcinoma (EOC), fallopian tube, and peritoneal carcinomas are grouped as a single clinical entity due to accumulating evidence of a common pathogenesis of these carcinomas, as well as their shared clinical manifestation and treatment [5].

Microtubule-interfering chemotherapeutic agents play an integral role in both the frontline and recurrent treatment of ovarian cancer. Paclitaxel and docetaxel both have unique microtubule-stabilizing and antiangiogenic properties that permit exceptional cytotoxic activity as chemotherapeutic agents. Paclitaxel (Taxol^®^, NSC 125973, Bristol-Myers Squibb, New York, NY, USA) was first shown to have antitumor activity in the 1960s as an extract from the bark of the Western or Pacific yew tree, *Taxus brevifolia* [6]. The C-13 ester side chain of paclitaxel is essential for antitumoral activity in that it binds to, promotes the assembly of, and stabilizes intracellular microtubules, which then inhibits depolymerization of these microtubules [6]. Docetaxel (Taxotere^®^, Sanofi-Aventis, Bridgewater, NJ, USA), paclitaxel’s semisynthetic analog derived from the bark of *Taxus baccata*, has similar cytotoxic activity due to its microtubule-stabilizing effect [6]. Other less frequently used microtubule-interfering cytotoxic agents that are also derived from plants include the vinca alkaloids (vinblastine, vincristine, and vinorelbine) and epipodophyllotoxins (etoposide and teniposide). Vinca alkaloids inhibit normal microtubular polymerization by binding to the tubulin subunit, which results in mitotic arrest by inhibiting the formation of the mitotic spindle [6]. Of the vinca alkaloids, vinorelbine has been the only one studied and used in the treatment of recurrent ovarian cancer as a single agent infusion [6]. Etoposide is produced as a gylcosidic derivative of podophyllotoxin by the mandrake plant (*Podophyllum peltatum)* and binds to the microtubular subunit tubulin, but the role of microtubular inhibition in microtubular assembly and the subsequent cytotoxicity is yet to be elucidated [7]. However, it is thought that the predominant cytotoxic effect stems from the interference with the normal functioning of topoisomerase II and promotion of single- and double-strand breaks in the DNA, resulting in cell death [7]. Etoposide, mostly in oral formulation, has been demonstrated to be effective in the treatment of recurrent platinum-resistant epithelial ovarian cancer [8]. Etoposide can be used as single agent or in combination therapy with other chemotherapy agents, including bevacizumab, irinotecan, and gemcitabine; however, further investigation is needed to determine the appropriate dosage and toxicity profile [8]. Epothilones are microtubule-stabilizing agents that are capable of inducing mitotic arrest, interfering with normal microtubule dynamics such as microtubule-dependent intracellular transport, ultimately resulting in cell death [9]. Ixabepilone (Ixempra^®^, Bristol-Myers Squibb, New York, NY, USA) is a semi-synthetic second-generation analog of epothilone B with cytotoxic activity that has been demonstrated in pre-treated ovarian cancer and may retain activity in taxane-treated patients, as it harbors several features that may overcome common mechanisms of taxane resistance [9].

Despite the development of new chemotherapeutic agents, carboplatin and paclitaxel remain the first line treatment in ovarian cancer. Nevertheless, paclitaxel and ixabepilone may have important roles as single-agent treatments for recurrent ovarian cancer. This review aims to describe the current and potential future therapeutic benefits of microtubule-interfering drugs in the treatment of epithelial ovarian cancer. Table 1a summarizes microtubule-interfering agents used in ovarian cancer treatment and Table 1b illustrates evidence for the use of microtubule-interfering agents in ovarian cancer treatment.

## 2. Role in Front-Line Treatment

### 2.1. Paclitaxel (Taxol^®^)

For most patients, EOC is treated surgically and followed by adjuvant platinum- and taxane-based chemotherapy. Alternatively, neoadjuvant chemotherapy (NACT) with the same cytotoxic agents prior to definitive surgery is another option in selected patients. Prior to the development of paclitaxel, cisplatin was typically given with cyclophosphamide. Early pharmacokinetic studies then demonstrated that paclitaxel and platinum compounds could be safely given together with little pharmacodynamic interactions [19]. Gynecologic Oncology Group (GOG) study 111 demonstrated that the use of paclitaxel 135 mg/m^2^ given over 24 h followed by cisplatin 75 mg/m^2^ was superior to cisplatin and cyclophosphamide (750 mg/m^2^) when each regimen was given every 3 weeks, with response rates of 73% versus 60%, respectively [20]. Nonetheless, the complete response rate was 51% compared to 36% in the paclitaxel and cyclophosphamide arms, respectively [20]. GOG 111 also showed a significant improvement in progression-free survival from 13 to 18 months and in overall survival from 24 to 38 months with the use of cisplatin and paclitaxel together [20]. Subsequent phase I trials with carboplatin in combination with paclitaxel demonstrated equivalent improved response rates albeit with a more favorable toxicity profile (i.e., reduced risk of emesis, neuropathy, nephropathy, and hearing loss) [21]. Piccart et al. later showed through a phase III randomized controlled trial paclitaxel’s superiority to cyclophosphamide in combination with platinum-based chemotherapy. The European Canadian intergroup trial (OV10) conducted by Piccart et al. demonstrated that paclitaxel when compared to cyclophosphamide improved overall (59% vs. 45%) and complete (41% vs. 27%) response rates, improved progression free survival (PFS) (15.5 months vs. 11.5 months), and improved overall survival (OS) (35.6 months vs. 25.8 months) [22].

However, the effectiveness of paclitaxel as a first line single chemotherapeutic agent for ovarian cancer was questioned when GOG 132 suggested cisplatin alone or in combination with paclitaxel had a superior response rate and PFS compared to paclitaxel alone [23]. These contradictory results were later thought to be due to the significant cross over between the different treatment arms prior to progression of disease. Nevertheless, the International Collaboration Ovarian Neoplasm 3 (ICON3) trial evaluated paclitaxel’s effect in combination with carboplatin in comparison to two control arms, which were carboplatin alone versus cisplatin, doxorubicin, and cyclophosphamide. There was no difference found in response rates, PFS, or OS between the three treatment groups [24]. Due to these results, as well as those of GOG 111 and OV10, the implementation of paclitaxel with a platinum drug became the preferred first line treatment for ovarian cancer [20,22]. To this day, the combination of carboplatin and paclitaxel continue to be the standard of treatment of epithelial ovarian cancer throughout the world and the regimen to which all other chemotherapeutic drugs are compared [22]. The role of dose-dense therapy (i.e., carboplatin administered every 3 weeks with paclitaxel administered weekly or both carboplatin and paclitaxel administered weekly) versus conventionally dosed therapy (i.e., carboplatin and paclitaxel administered every 3 weeks) continues to be debated. Overall, randomized trials have suggested equivalent or improved efficacy with dose-dense regimens relative to conventionally dosed therapy, though toxicities are usually higher with the dose-dense regimens [25,26,27,28,29,30]. In some cases, dose-dense chemotherapy has been shown to be well tolerated and could be considered as frontline treatment in advanced ovarian cancer [31,32]. In patients with optimally resected stage III EOC, GOG 158 demonstrated that carboplatin with paclitaxel is less toxic, easier to administer, and not inferior to cisplatin plus paclitaxel [33].

### 2.2. Docetaxel (Taxotere^®^)

Once the importance of paclitaxel as a cytotoxic drug was appreciated, the Western or Pacific yew tree from which it was derived was harvested at an unsustainable rate. Therefore, researchers discovered a method to synthesize a semisynthetic form of paclitaxel from the needles of the European yew tree, which they called docetaxel [34]. A large randomized controlled trial, the Scottish Randomised Trial in Ovarian Cancer (SCOTROC), compared the response rates and toxicity profiles of docetaxel to those of paclitaxel. The SCOTROC phase III trial included over 1000 patients and demonstrated relatively identical efficacy between the two treatments. As such, the OS rates (64% and 68%) and tumor response rates (58% and 59%) were seen with docetaxel and paclitaxel treatments, respectively. The major differences between the two treatments were their toxicity profiles, with grade 3 and 4 neutropenia being more common with docetaxel and neuropathy more common with paclitaxel [35,36]. Nonetheless, docetaxel can be administered along with carboplatin, but paclitaxel is still often preferred due to the fact that it is less myelosuppressive than docetaxel. However, a consideration between treatment with these two taxanes may be individualized based on their differing toxicity profiles [35]. Docetaxel carries a higher risk for neutropenia, hypersensitivity reactions, and nausea and emesis, whereas paclitaxel administration includes a greater risk of neuropathy, weakness, and myalgias. Some have suggested that the difference in the toxicity profiles of these otherwise similar drugs could be attributed to longer periods of docetaxel retention within cells [37].

### 2.3. Nab-Paclitaxel (Abraxane^®^)

Paclitaxel and docetaxel are undoubtedly the most widely used cytotoxic taxanes in the treatment of ovarian cancer. Premedication is routinely recommended in order to prevent infusion reactions during the administration of these agents. In early trials with paclitaxel, the incidence of infusion reactions was as high as 30%, but after the introduction of methods to diminish the incidence and severity of these reactions (i.e., premedication with antihistamines and glucocorticoids, prolongation of infusion time, etc.), the rate of severe reaction has been dramatically reduced to a mere 2–4% [38,39,40,41]. Nowadays, with the appropriate premedication regimen, the incidence of infusion reactions is the same whether paclitaxel is administered over 1, 3, or 24 h, but the incidence may be higher in infusion times under 1 h [11,42,43]. It was thought that due to differences in the formulation of docetaxel, the rate of infusion reactions would be less than paclitaxel; however, an equivalent percentage of patients receiving docetaxel without premedication develop infusion reactions [44]. Despite standard premedication prior to docetaxel administration, approximately 2% of patients may have a potentially life-threatening infusion reaction [45]. There is evidence that both the taxane component and the vehicles used to solubilize these agents are capable of causing infusion reactions. For instance, paclitaxel is formulated in Cremophor^®^ (BASF, Florham Park, NJ, USA) and docetaxel is formulated in a vehicle called polysorbate 80; however, the drugs themselves can be capable of initiating infusion reactions independent of their solvents [46,47]. For those patients who cannot receive paclitaxel as a result of experiencing an infusion reaction, an alternative treatment option exists, which is nanoparticle albumin-bound paclitaxel, also known as nab-paclitaxel. Nab-paclitaxel obviates the need for Cremophor^®^ (BASF, Florham Park, NJ, USA) as the vehicle and infusion reactions have not been seen in phase I, II, or III studies [48,49,50].

The GOG conducted a phase II evaluation of nab-paclitaxel in the treatment of recurrent or persistent platinum-resistant ovarian cancer, where nab-paclitaxel was shown to have a favorable efficacy and toxicity characteristics relative to other solvent-based taxanes, such as paclitaxel and docetaxel [51]. While the overall toxicity profile of nab-paclitaxel is better as compared with conventional paclitaxel, the incidence of transient sensory neurotoxicity may be slightly higher. This was illustrated by a phase III randomized trial comparing nab-paclitaxel 260 mg/m^2^ every three weeks to standard paclitaxel 175 mg/m^2^ every three weeks [50]. In this study, nab-paclitaxel was associated with a higher rate of grade 3 sensory neuropathy (10% versus 2%); however, the paclitaxel dose was 49% higher in the nab-paclitaxel arm, a fact that could easily have accounted for the higher frequency of neuropathy. Nonetheless, grade 3 neuropathy is reversible to grade ≤1 in approximately 50% of patients, and most have clinical improvement in symptoms within one month of treatment discontinuation [52].

## 3. Role in Recurrent Disease

The management of recurrent ovarian cancer is stratified based on the amount of time that has elapsed between the completion of platinum-based treatment and the detection of relapse. This time frame is known as the platinum-free interval (PFI). Patients with a PFI of six months or longer are considered to have platinum-sensitive disease, whereas patients with a PFI of less than six months are considered to have platinum-resistant disease. Initial treatment with carboplatin and paclitaxel leads to remission in 60% to 80% of patients [6]. Unfortunately, the overall likelihood of relapse after initial therapy for all stages of disease is 62%, and up to 80–85% for women who present with advanced stage disease [53]. Phase III trials have shown the superiority of combination platinum-based chemotherapy in the management of platinum sensitive ovarian cancer compared with single-agent therapy [10,12,54]. These platinum-based regimens include carboplatin/paclitaxel, carboplatin/gemcitabine, and carboplatin/pegylated liposomal doxorubicin [12,54]. Of these, paclitaxel is the only agent that has shown an OS benefit when combined with platinum-based chemotherapy in a clinical trial [12]. The importance of paclitaxel use in recurrent ovarian cancer was demonstrated in the ICON4/Arbeitsgemeinschaft Gynakologische Onkologic-Ovary 2.2 trial (AGO-OVAR-2.2), where the use of platinum chemotherapy with and without concomitant paclitaxel in recurrent ovarian cancer was studied [54]. This study demonstrated that the use of platinum and paclitaxel together significantly improved progression-free survival by 10% or median of 3 months [54]. Furthermore, the addition of bevacizumab, the vascular endothelial growth factor receptor (VEGFR) antibody, to combination platinum-based chemotherapy has been shown to improve the objective response rate (ORR) and prolong the time to subsequent disease progression compared to chemotherapy alone [12].

For patients with recurrent platinum-resistant ovarian cancer, there are a number of single-agent cytotoxic drugs that have been shown to have a modest ORR (10–15%) including ixabepilone [9] and patupilone [10] and pegylated liposomal doxorubicin [10]; however, there remain no data to suggest the superiority of one single agent over another. Oftentimes, paclitaxel is the preferred single-agent option, especially in those who have not been previously treated with paclitaxel for recurrent disease if prior toxicities do not prohibit its repeated use (i.e., persistent neuropathy or prior prolonged myelosuppression) [8]. Paclitaxel administered on a weekly dose dense regimen has been suggested to be efficacious in those with platinum and paclitaxel resistant ovarian cancer. In fact, response rates to dose dense weekly paclitaxel of 20.9% were seen in patients with ovarian cancer resistant to both platinum and paclitaxel, while response rates were as high as 60% in those with platinum-resistant ovarian cancer [13,15].

For patients with platinum-resistant ovarian cancer, combining single-agent chemotherapy with bevacizumab may improve objective ORR and PFS compared to cytotoxic therapy alone. For those who received weekly paclitaxel with bevacizumab the ORR was 53% versus 30% and the PFS was 10 months versus 4 months with or without bevacizumab, respectively. Similar improvements in ORR and PFS were seen with the addition of bevacizumab to single-agent topotecan and pegylated liposomal doxorubicin [14]. Unfortunately, the acquisition of progressive chemoresistance with each recurrence eventually results in death. This has fueled the search for additional effective therapeutic agents. Epothilones are microtubule-stabilizing agents that are capable of inducing mitotic arrest, interfering with normal microtubule dynamics, such as microtubule-dependent intracellular transport, resulting in cell death. Epothilones are less susceptible than taxanes to overexpression of P-gylcoprotein, the presence of certain tubulin isoforms (class III β-tubulin), and tubulin mutations, which have been implicated in taxane resistance [16,55,56]. Ixabepilone (Ixempra^®^) is a semi-synthetic second-generation analog of epothilone B with cytotoxic activity that has been demonstrated in pre-treated ovarian cancer and may retain activity in taxane-treated patients by virtue of several features that may overcome common mechanisms of taxane resistance [9]. In GOG 126M, a phase II study of ixabepilone 20 mg/m^2^ days 1, 8, and 15 every 28 days in platinum/taxane-resistant ovarian cancer, ORR was 14.3% with a 4.4-month time to progression and median OS of 14.8 months [9]. Overall, ixabepilone demonstrated antitumor activity as well as an acceptable safety profile with adverse effects including peripheral grade 2 (28.5%) and grade 3 (6.1%) neuropathy, grades 3 to 4 neutropenia (20.4%), grade 3 fatigue (14.3%), grade 3 nausea/emesis (22%), grade 3 diarrhea (10%), and grade 3 mucositis (4%) [9].

In a phase III study of patupilone (epothilone B) versus pegylated liposomal doxorubicin in platinum-resistant or -refractory ovarian cancer, ORR was higher in the patupilone arm (15.5% vs. 7.9%; odds ratio, 2.11; 95% CI, 1.36 to 3.29), though both agents achieved similar PFS and OS [10]. Observed adverse events (AEs) of any grade in the patupilone arm included diarrhea (85.3%) and peripheral neuropathy (39.3%), whereas in the pegylated liposomal doxorubicin arm, mucositis/stomatitis (43%) and hand-foot syndrome (41.8%) were the most commonly observed AEs [10]. There are also encouraging retrospective data demonstrating efficacy of ixabepilone with bevacizumab in patients with platinum/taxane-resistant ovarian cancers [57]. A subsequent randomized phase II investigation demonstrated that the combination of ixabepilone with bevacizumab resulted in an ORR of 33% with a PFS of 5.5 months and median OS of 10.0 months. Prior treatment with bevacizumab did not influence PFS or OS. The population was heavily pre-treated, with 51% of patients in receipt of >3 prior lines and 18% platinum-refractory [17]. Tumor expression of class III β-tubulin by immunohistochemistry did not predict response, underscoring the need for improved predictive biomarkers of response to microtubule-stabilizing agents [17]. Therefore, epothilones (ixabepilone and patupilone) may retain antitumor activity in taxane-treated patients, since they harbor several features that may overcome mechanisms of taxane resistance while having an overall tolerable side effect profile.

## 4. Future Directions

### 4.1. Understanding Mechanisms of Resistance to Microtubule-Interfering Agents

Microtubules consist of α/β tubulin heterodimers that exhibit dynamic instability, the ability to shorten and lengthen during critical cellular processes including cell division. The rapidly polymerizing ‘plus-end’ is capped by β tubulin, whereas the less dynamic ‘minus end’ is capped by α-tubulin. Guanosine 5-triphosphate (GTP) must be present for polymerization and is subsequently hydrolyzed. Paclitaxel and epothilones share overlapping binding sites on β-tubulin [18]. While taxanes and epothilones exert their main effect by hyper-stabilization of microtubules resulting in mitotic arrest, paclitaxel may also induce apoptosis through damage to microtubules in interphase, induction of cdc-2 kinase during metaphase/anaphase, phosphorylation of BCL-2, and induction of IL-1 ß and TNF [58]. Well-described mechanisms of paclitaxel resistance [59] include increased drug efflux via p-glycoprotein (*ABCB1*; *MDR-1*) gene amplification [60], increased transcription/translation [61], or mutations that enhance pump function [62]. Mutations in β-tubulin or altered isotype expression may underlie reduced drug binding affinity [63]. In vertebrates, β-tubulin exists in at least eight isotypes, with various functions and distributions [64]. Class III β-tubulin overexpression is one of the most extensively studied for its role in chemoresistance, but clinical applications remain investigational only [65]. Microtubule-stabilizing proteins (MAPs), such as tau and MAP2, may affect paclitaxel binding [66], but studies of its practical use in ovarian cancer are conflicting [67,68]. MAP4, the plus-end binding protein EB-1, CLIP-170, MCAK, and stathmin regulate microtubule dynamics and may modulate sensitivity to taxanes and epothilones [69,70]. A number of post-translational modifications (PTMs) of α-tubulin govern the complexity of microtubule regulation, including acetylation, tyrosination, and polyglycylation; β-tubulin undergoes phosphorylation and polyglutamylation [71]. A role for such PTMs in chemoresistance is emerging [72]. Alteration of the drug target, enhanced metabolism, deranged nuclear-cytoplasmic shuttling, and increased ability to counter drug-induced damage or apoptosis also contribute to drug resistance. A summary of several mechanisms of resistance to microtubule-interfering agents is provided in Figure 1 [73,74,75]. Numerous potential biomarkers of response to microtubule-active agents have been proposed, including class I beta-tubulin somatic mutations and differential expression of parkin, survivin, Aurora A kinase, Bcl-2, galectin-1 [76], HE4 [77], MAD1 [78], PRP4K [79], and ERCC1 [67,80,81].

Microtentacles, first described in 2007 in detached apoptotic-resistant breast cancer cells, are tubulin-based projections that may modulate invasion and are distinct from the actin-based protrusions that govern two-dimensional (i.e., lamellipodia, filopodia, and blebs) and three-dimensional (invadopodia or podosomes) cell movement [82,83,84,85,86,87]. In addition to breast cancer, microtentacles have been observed in glioblastoma [88]. Recently, we have also identified microtentacles in ovarian cancer cells isolated from ascites and characterized their formation and dynamicity in relationship to the tubulin isotype and chemoresistance to microtubule-stabilizing and de-stabilizing agents [89] (see accompanying article in this issue). Microtentacles may be a particularly relevant mechanism of metastases and chemoresistance in ovarian cancer compared to other solid tumors due to the hallmark formation of ascites and pattern of intra-peritoneal spread at the time of diagnosis and recurrence. Furthermore, intra-abdominal pressure and fluid shear stresses from the accumulation and circulation of ascites likely induce epithelial to mesenchymal transition (EMT) and influence adhesion and migration in these floating tumor cells [90]. Single-cell tethering techniques to visualize microtentacles after drug exposure may even allow rapid identification of drug resistance without cell culture or xenografts [91].

### 4.2. Mitigating the Limitations and Toxicities of Microtubule-Interfering Agents

#### 4.2.1. Drug Resistance

Microtubule active agents can bind one of four locations within the microtubule: the taxoid site (inner surface of the beta subunit), the vinca site (tip of microtubule), the colchicine site (copolymerization with the microtubule causing steric curvature of beta tubulin), and the laulimalide-peloruside site (exterior of the beta subunit) [92]. Knowledge of the binding properties of these sites can allow for rational drug design to overcome drug resistance.

##### Alternative Microtubule Stabilizing Agents

Numerous taxoid (e.g., Larotaxel [XRP9881, Sanofi], BMS-184476, TPI-287) and epothilone (e.g., KOS-1584) derivatives have been studied for their promise in taxane-resistant disease and lack of participation in p-glycoprotein drug export [93,94,95,96].

Many non-taxane microtubule-interfering agents also remain under development. Taccalonolides are pentacyclic steroids discovered in 1963 as isolates from *Tacca plantaginerea Andrea* [97]. These compounds appear not to share cross-resistance with paclitaxel and stabilize microtubules via a unique mechanism with defective β-tubulin spindles and interphase microtubule bundling yielding G2/M arrest [98]. Cyclostreptin (FR182877, Fujisawa Pharmaceutical Co, Tokyo, Japan) was generated from *Streptomyces* in 1998 and is particularly effective at stabilizing microtubules at low temperatures, and may additionally promote tubulin polymerization [96,97,99,100]. Dictyostatin [101], dictyostatin-1, discodermolide [102], eleutherobin/sarcodictyins A & B [103], and zampanolide are isolated from marine organisms [104,105]; these compounds share an overlapping binding site with paclitaxel on microtubules and exhibit similar mechanisms of action, yet retain activity in paclitaxel-and epothilone-resistant cells. Fully synthetic compounds that bind the paclitel pocket include GS-164, synstab A, 4′-methoxy-2-styrylchromone, dienone derivatives, aromatic ketones, pyranochalcone derivatives, alpha-cyano-bis(indolyl)chalcones, and cyclopropylamide analogs of combretastatin A4 [94].

Laulimalide, peloruside, and ceratamines are also derived from marine sponges but bind tubulin at a site distinct from paclitaxel on the exterior of the microtubule to promote microtubule stabilization [106]. The clinical utility of laulimalide may be limited by toxicity [107]. Peloruside A is a marine-derived macrolide that serves as a competitive inhibitor of laulimalide [108]. It acts synergistically with agents that bind the taxoid site and has improved water solubility relative to paclitaxel [109]. Compared to laulimalide and peroruside, the structure of ceratamines is much less complex, making it synthetically attractive [110].

##### Alternative Microtubule-Destabilizing Agents

Eribulin mesylate (Halaven^®^, Eisai, Tokyo, Japan) binds near the vinca domain and is FDA-approved for use in breast cancer (2010) after failure of two prior regimens for late-stage disease and unresectable liposarcoma (2016) following treatment with an anthracycline. It is a synthetic analog of halichondrin B derived from the marine sponge *Halichondria okadai*. Its mechanism of action involves inhibition of the plus ends of the microtubule [111,112]. In a phase II study of ovarian cancer patients, eribulin produced an objective response rate of 5.5% with median progression-free survival of 1.8 months [113]. Vinflunine (Javlor^®^, BMS, Brooklyn, NY, USA) was the first clinically relevant fluorinated vinca alkaloid. It has approval in Europe (2009) for treatment of metastatic or advanced urothelial carcinoma after failure of platinum. It has the weakest affinity among vinca derivatives for tubulin and inhibits treadmilling less impressively than vinorelbine or vinblastine, but has notable antivascular effects [114]. Many microtubule depolymerization agents inherently harbor vascular disrupting properties at the vascular endothelial junction, with tumor selectivity possibly due to the fragility of tumor vessels [115]. No clinical trials have been conducted towards use of vinfkuninee in ovarian cancers. Combretastatins (e.g., combretastatin A-1 [CA1], Oxi4503, Fosbretatubulin, and Ombrabulin; reviewed in Borys et al., 2021), extractions of the *Combretum caffrum* tree, target the cochicine binding site of tubulin, but their development has been largely suspended due to underwhelming efficacy [116]. There is currently interest in development of photopharmacologic derivatives of combretastatins to enhance cytotoxicity, sometimes hybridized to additional cytotoxic agents [117]. Celogentin/moroidin peptides isolated from the seed of *Celosia argentea* have an inhibitory potency equal to or greater than vinblastine [118]. Cryptophycins isolated from cyanobacteria exert extremely potent effects on tubulin depolymerization via the vinca binding domain; notably, the cryptophycin LY573636 was recently studied at the phase II level for second to fourth line therapy for platinum-resistance ovarian cancer with a response rate of 12% [119].

#### 4.2.2. Water Insolubility

Water insolubility impairs penetration of paclitaxel across the blood–brain barrier. Paclitaxel diluents also frequently prompt hypersensitivity reactions. Cabazitaxel (Jevtana^®^, previously known as XRP6258, TXD258, and RPR116258A, Sanofi, Paris, France) is a modification of docetaxel that may have improved activity in resistant disease and better penetration of the blood–brain barrier; it has been approved for metastatic prostate cancer since 2010 [120]. Paclitaxel poliglumex/CT-2103 (Opaxio^®^, Cell Therapeutics Inc., Seattle, WA, USA) is a water-soluble polymer of glutamic acid linked to paclitaxel. The agent was studied in GOG212 but failed to improve ovarian cancer survival in the maintenance setting [121]. Docosahexaenoic acid–paclitaxel (DHA–paclitaxel, Taxoprexin^®^, Protarga, King of Prussia, PA, USA) consists of covalently conjugated essential fatty acids to the 2′-OH position of the paclitaxel molecule; despite encouraging pre-clinical activity, it failed to demonstrate any survival benefit over dacarbazine in melanoma [122]. EndoTAG-1^®^ (MediGene, Melbourne, VIC, Australia) contains cationic liposomes and paclitaxel, with specific anti-angiogenic activity; it remains under investigation in the phase III setting for pancreatic cancer [123]. Polymeric micellar paclitaxel (Genexol-PM^®^, Samyang Genex Co., Gyeonggi-do, Korea; Shanghai Yizhong Biotechnical Co., Ltd. Shanghai, China) has recently demonstrated improved response rates and survival compared to paclitaxel in non-small cell lung cancers [124].

Several oral formulations of paclitaxel and docetaxel have been developed to circumvent hypersensitivity reactions brought upon by hydrophobic diluents [125,126]. These include DHP107, a mixture of paclitaxel with monoolein/tricarprylin/Tween^®^ 80 (DAE HWA Pharmaceutical Co., Ltd., Seoul, Republic of Korea), ModraDoc001, an amalgamation of docetaxel with polyvinylpyrrolidone-K30 and sodium lauryl sulphate [127], Orataxel (IDN-5109, BAY 59-8862) [128], Milataxel (MAC-321) [129], BMS-275183 [130], and tesetaxel (DJ-927, Genta, LaJolla, CA, USA) [131], among others. Compounds T13 and T26 have improved oral bioavailability in rats [132]. Unfortunately, the oral bioavailability of these agents in human clinical trials appears to vary significantly.

#### 4.2.3. Peripheral Neuropathy, a Dose-Limiting Toxicity of Microtubule-Interfering Agents

After eight cycles of carboplatin AUC 6 and paclitaxel 175 mg/m^2^ every three weeks employed in front-line treatment of ovarian cancer, 36% of patients ≥70 years and 20% of patients <70 years will experience peripheral neuropathy of grade 2 or greater [133]. After six cycles of carboplatin and paclitaxel in a front-line setting, 15% of patients will have residual neuropathy at 6 months and 11% of patients at 2 years. Docetaexl is less neurotoxic than paclitaxel. Chemotherapy-induced peripheral neuropathy produces numbness, hyperalgesia, tingling, and changes in proprioception in a stocking-glove distribution [134]. Rarely, motor neuropathy can occur. Autonomic neuropathy with vinca alkaloids can cause orthostatic hypotension and constipation [135]. The likelihood of symptoms is cumulative with onset, sometimes months after exposure.

The mechanism of chemotherapy-induced neuropathy may relate to aberrant calcium signaling and biochemical and structural aberrations of nervous system mitochondria, dorsal root ganglia, glia, and astrocytes [136]. Risk factors for the development of chemotherapy-induced neuropathy include older age, prior exposure to neurotoxins, diabetes, folate/B12 deficiencies, inter-individual variations in CYP2C8*3 [137], class IIa β-tubulin [138], FDG4, FZD3, EPHA5 [139], and the WNT pathway, among others [140].

Approaches to the prevention and treatment of neuropathy induced by microtubule-stabilizing agents are required. Amifostine, an antioxidant, has shown inconsistent benefit in prevention of taxane- and platinum-induced neuropathy, at the expense of potential nausea, lightheadedness, cardiovascular, and dermatologic adverse events, and its use is not currently endorsed by the American Society of Clinical Oncology [141]. Vitamin E, recombinant human leukemia inhibitory factor, retinoic acid, glutathione, and amitriptyline have no proven benefit [135,142]. A 5-week course of duloxetine, a serotonin/norepinephrine re-uptake inhibitor, reduces pain in patients with grade 1 or higher sensory neuropathy, but is more effective for platinum-induced rather than taxane-induced neuropathy [143]. Gabapentin exerts selective inhibition of voltage-gated calcium channels to decrease post-synaptic excitatory neurotransmitters and has been shown to be effective in other forms of neuropathy [144]. Lithium prevents paclitaxel-induced neuropathy in mice without diminishing anti-cancer efficacy [145]. Studies that examine nicotinamide riboside (NCT04112641), glucosides and rutinosides (NCT04669977), stretching (NCT03272919), transcranial current stimulation (NCT04107272; NCT04833920), vibration (NCT04959929), cryoompression (NCT05095051), and metformin (NCT04780854) for prevention and treatment of chemotherapy-induced peripheral neuropathy are currently recruiting.

## 5. Ongoing Clinical Trials Involving Microtubule Inhibitors

A search of the U.S. National Library of Medicine clinicaltrials.gov with key words “microtubules” and “ovarian cancer” yielded five ongoing clinical trials involving microtubule inhibitors in ovarian cancer treatment. Dr. Fu and colleagues at MD Anderson Cancer Center are recruiting for their phase 1 clinical trial titled “First-in-Human Evaluation of GRN-300 in Subjects with Recurrent Ovarian, Primary Peritoneal, and Fallopian Tube Cancers” (NCT04711161). They plan to compare GRN-300 (a salt-inducible kinase inhibitor) as monotherapy versus in combination with paclitaxel. Outcomes of interest include dose determination and toxicity.

In a multicenter trial sponsored by AstraZeneca, investigators will be comparing the role of AZD5305 monotherapy versus combination with paclitaxel versus combination with carboplatin +/− paclitaxel in treatment of advanced ovarian cancer among other solid tumors (NCT04644068). Their study is titled “Study of AZD5305 as Monotherapy and in Combination with Anti-cancer Agents in Patients With Advanced Solid Malignancies (PETRA)”.

The third ongoing clinical trial is jointly sponsored by ImmunoGen, Inc., the Gynecologic Oncology Group, and the European Network of Gynaecological Oncological Trial Groups in a trial titled “A Study of Mirvetuximab Soravtansine vs. Investigator’s Choice of Chemotherapy in Platinum-Resistant, Advanced High-Grade Epithelial Ovarian, Primary Peritoneal, or Fallopian Tube Cancers with High Folate Receptor-Alpha Expression” (NCT04209855). The investigators’ choice of chemotherapy includes paclitaxel or topotecan or Pegylated liposomal doxorubicin.

Next is ATLAS-101, a Phase I/II, Dose Escalation and Dose Expansion study involving AMXI-5001 in treatment of advanced malignancies including ovarian cancers (NCT04503265). AMXI-5001 is a dual PARP (poly adenosine diphosphate [ADP] ribose polymerase) and microtubule polymerization inhibitor (NCT04503265).

Lastly, “A Study of RGX-104 in Patients with Advanced Solid Malignancies and Lymphoma” is a Phase 1 dose escalation and expansion study of RGX-104, an oral small molecule targeting the liver X receptor (LXR), as a single agent and in combination with nivolumab, ipilimumab, docetaxel, or pembrolizumab plus carboplatin/pemetrexed (NCT02922764). Patients with persistent of recurrent epithelial ovarian cancer are eligible for the study.

## 6. Conclusions

Microtubule-interfering drugs are well tolerated and for many years have been the backbone of the standard treatment in both the front line and recurrent setting of ovarian cancer. Acquisition of progressive chemoresistance with each recurrence eventually results in death. Epothilones (ixabepilone and patupilone) may retain activity in taxane-treated patients, since they harbor several features that may overcome mechanisms of taxane resistance while having an overall tolerable side effect profile. Therefore, agents such as epothilones strengthen the armamentarium of limited treatment options for recurrent/refractory epithelial ovarian cancer. Undoubtedly, there is still an enormous unmet medical need for novel therapeutic agents that can be used to treat patients with epithelial ovarian cancer that portends such a poor prognosis. Nevertheless, ongoing clinical trials will shed further light into the potential of incorporating new microtubule-interfering drugs as part of current standard practice.

## Figures and Tables

**Figure 1 cancers-13-06239-f001:**
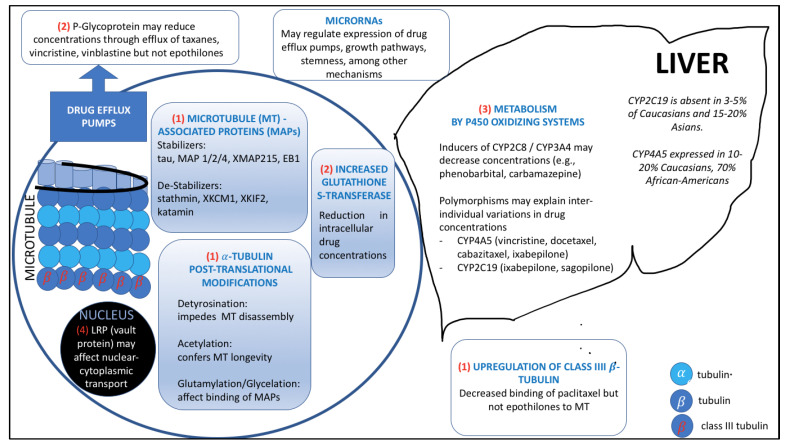
Mechanisms of drug resistance include alteration of the drug target (1), reduction of concentrations due to enhanced export from the cell (2) or enhanced metabolism (3), altered nuclear-cytoplasmic shuttling (4), and increased ability to counter drug-induced damage or apoptosis through altered expression of Bax, Bcl-2, AKT, surviving, sorcin, MDA, SKP2, F-Box proteins, Gli proteins, and sphingolipids, among many others.

**Table 1 cancers-13-06239-t001:** (**a**) Microtubule-interfering cytotoxic agents used in the treatment of ovarian cancer. (**b**) Evidence for microtubule-interfering cytotoxic agents used in frontline and recurrence treatment of ovarian cancer.

**(a)**						
**Taxanes**		**Origin**	**Use**	**Mechanism of Action**	**Resistance**	**Toxicity**
Paclitaxel (Taxol ^®^) [6]		Bark of the Western or Pacific yew tree, *T. brevifolia*	First or second line ovarian cancer	Stabilizes and inhibits depolymerization of intracellular microtubules	Overexpression of multidrug resistance (MDR-1) gene; molecular changes in the target molecule (betatubulin); changes in checkpoint proteins; changes in lipid composition and overexpression of interleukin 6 (IL-6)	Neuropathy, weakness, myalgias, myelosuppression
Docetaxel (Taxotere^®^) [6]		Semisynthetic analog derived from the bark of *T. baccata*	Refractory ovarian cancer	Cytotoxic activity through microtubule-stabilization	Limits intracellular drug concentration and stabilization; inhibits cytotoxic effects through alternative growth pathways or apoptotic escape	Neutropenia, hypersensitivity reactions, nausea, emesis
**Vinca alkaloids**		**Origin**	**Use**	**Mechanism of Action**	**Resistance**	**Toxicity**
Vinblastine [6]		Madagascar periwinkle	Advanced ovarian cancer	Prevents polymerization and assembly of microtubules; disrupts mitotic spindle and cytoskeletal function	Enhanced efflux via P-glycoprotein in the cell membrane	Phlebitis, cellulitis, nausea, vomiting, diarrhea, alopecia, myelosuppression, SIADH
Vincristine [6]		Madagascar periwinkle *Catharanthus roseus*	Used in combination with other agents	Microtubule destabilizing antimitotic activity	Overexpression of efflux pumps and tubulin isotypes; modifications of the target microtubules	Alopecia, GI symptoms, neuropathy, weight loss
Vinorelbine [6]		Semi-synthetic	As a single agent in recurrent ovarian cancer	Inhibits mitotic spindle formation and microtubule polymerization causing mitotic arrest	Modification of transport system	Peripheral neuropathy, anemia, hyponatremia, GI symptoms, phlebitis.
**Epipodophyllotoxins**		**Origin**	**Use**	**Mechanism of Action**	**Resistance**	**Toxicity**
Etoposide [7]		Gylcosidic derivative of podophyllotoxin by the mandrake plant (*P. peltatum)*	Recurrent platinum-resistant epithelial ovarian cancer	Interferes with topoisomerase II function and promotes single- and double-strand DNA breaks, resulting in cell death	Altered expression of topoisomerase II; multidrug-resistant phenotypes encoded by the mdr1 and MRP (multidrug resistance-associated protein) genes.	Myelosuppression, mucositis, nausea, alopecia, emesis
Teniposide [7]		Semisynthetic derivative of podophyllotoxin from the mandrake plant (*P. peltatum*)	Advanced refractory ovarian cancer	Inhibits topoisomerase II activity; prevents cell mitosis by causing single and double stranded DNA breaks and protein cross linking	Altered expression of topoisomerase II, and the multidrug-resistant phenotypes encoded by the mdr1 and MRP genes.	Bone marrow suppression, gastrointestinal toxicity, hypersensitivity reactions, reversible alopecia
**Epothilones**		**Origin**	**Use**	**Mechanism of Action**	**Resistance**	**Toxicity**
Ixabepilone (Ixempra^®^) [9]		Semi-synthetic second-generation analog of epothilone B	Platinum-resistant or refractory ovarian cancer.	Induces cell death by interfering with microtubule function such as intracellular transport.	Increased βIII-tubulin expression. Mutations in β274Thr→Ile and β282Arg→Gln resuting in impaired abiity to induce tubulin polymerization	Neutropenia, peripheral neuropathy
Patupilone (epothilone B) [10]		Myxobacterium *Sorangium cellulosum*	Paclitaxel-resistant ovarian cancer	Induces cell-cycle arrest and apoptosis by binding to B-tubulin	Mutations in β274Thr→Ile and β282Arg→Gln resuting in impaired abiity to induce tubulin polymerization	Diarrhea, peripheral neuropathy, fatigue
**(b)**						
**Year**	**Authors**	**Title**	**Number of Patients Enrolled**	**Treatment Arms**	**Clinical Outcomes**	**Toxicity**
1994 *	Eisenhauer et al. [11]	European-Canadian randomized trial of paclitaxel in relapsed ovarian cancer: high-dose versus low-dose and long versus short infusion.	382	Randomized in a bifactorial design to receive either 175 or 135 mg/m^2^ of Taxol over either 24 or 3 h.	Response was slightly higher at the 175-mg/m^2^ dose than at 135 mg/m^2^ (20% vs. 15%; *p* = 0.2). PFS was significantly longer in the high-dose group (19 vs. 14 weeks; *p* = 0.02). ORR were similar in the 24- and 3-h groups (19% and 16%, respectively; *p* = 0.6). No survival differences were noted.	24-h taxol infusion was associated with significantly more neutropenia.
2003	Parmar et al., ICON4/AGO-OVAR-2.2 trial [12]	Paclitaxel plus platinum-based chemotherapy versus conventional platinum-based chemotherapy in women with relapsed ovarian cancer: the ICON4/AGO-OVAR-2.2 trial	802	Paclitaxel plus platinum chemotherapy or conventional platinum-based chemotherapy.	Paclitaxel plus platinum was associated with longer 2-year survival (57% vs. 50%) and 1-year PFS (50% vs. 40%).	Paclitaxel plus platinum was associated with more alopecia and neurotoxicity.Conventional platinum-based chemotherapy was associated with myelosuppression.
2006	Markman et al., GOG-126 N [13]	Phase II trial of weekly paclitaxel (80 mg/m^2^) in platinum and paclitaxel-resistant ovarian and primary peritoneal cancers: a Gynecologic Oncology Group study	48	Patients with platinum- and paclitaxel-resistant ovarian cancer (defined as progression during, or recurrence < 6 months following, their prior treatment with both agents) received single agent weekly paclitaxel (80 mg/m^2^/week) until disease progression (assuming acceptable toxicity).	Weekly administration of paclitaxel can be useful in women with both platinum- and paclitaxel-resistant ovarian cancer. The ORR was 20.9%.	Serious adverse events were relatively uncommon (neuropathy-grade 2: 21%; grade 3: 4%; and grade 3 fatigue: 8%).
2009 *	Sharma et al. [14]	Extended weekly dose-dense paclitaxel/carboplatin is feasible and active in heavily pre-treated platinum-resistant recurrent ovarian cancer.	20	Patients with platinum-resistant/refractory ovarian cancer received carboplatin AUC 3 and paclitaxel 70 mg/m(^2^) on day 1, 8, and 15 every 4 weekly for six planned cycles.	Response rate was 60% by radiological criteria (RECIST) and 76% by CA125 assessment. Median PFS was 7.9 months and OS was 13.3 months.	Grade 3 toxicities consisted of neutropenia (29% of patients) and anemia (5%).
2010	De Geest et al., GOG-0126M, NCT00025155 [9]	Phase II Clinical Trial of Ixabepilone in Patients With Recurrent or Persistent Platinum- and Taxane-Resistant Ovarian or Primary Peritoneal Cancer: A Gynecologic Oncology Group Study	49	Intravenous ixabepilone 20 mg/m^2^ administered over 1 hour on days 1, 8, and 15 of a 28-day cycle.	The ORR was 14.3%, with median PFS of 4.4 months. SD was achieved in 40.8% of patients. Ixabepilone seems to be an active cytotoxic agent in patients with recurrent platinum- and taxane-resistant ovarian or primary peritoneal carcinoma.	Adverse effects included peripheral neuropathy, neutropenia, fatigue, nausea/emesis, diarrhea, and mucositis.
2012	Colombo et al., NCT00262990 [15]	Randomized, open-label, phase III study comparing patupilone (EPO906) with pegylated liposomal doxorubicin in platinum-refractory or -resistant patients with recurrent epithelial ovarian, primary fallopian tube, or primary peritoneal cancer.	829	Patients were randomly assigned to receive patupilone 10 mg/m^2^ IV every 3 weeks or pegylated liposomal doxorubicin (PLD) 50 mg/m^2^ IV every 4 weeks.	There was no statistically significant difference in OS between the patupilone and PLD arms (13.2 and 12.7 months respectively, *p* = 0.195). Median PFS was 3.7 months for both arms. The ORR was higher in the patupilone arm than in the PLD arm (15.5% vs. 7.9%).	Frequently observed adverse events included diarrhea (85.3%) and peripheral neuropathy (39.3%) in the patupilone arm and mucositis/stomatitis (43%) and hand-foot syndrome (41.8%) in the PLD arm.
2014	Pujade-Lauraine et al. [16]	Bevacizumab combined with chemotherapy for platinum-resistant recurrent ovarian cancer: The AURELIA open-label randomized phase III trial.	361	Randomized to single-agent chemotherapy alone (PLD, weekly paclitaxel, or topotecan) or with bevacizumab (10 mg/kg every 2 weeks or 15 mg/kg every 3 weeks) until progression, unacceptable toxicity, or consent withdrawal.	Median PFS was 3.4 months with chemotherapy alone versus 6.7 months with bevacizumab-containing therapy. The OS was 13.3 vs. 16.6 months, respectively.	Hypertension and proteinuria were more common with bevacizumab. GI perforation occurred in 2.2% of bevacizumab-treated patients.
2015	Roque et al. [17]	Weekly ixabepilone with or without biweekly bevacizumab in the treatment of recurrent or persistent uterine and ovarian/primary peritoneal/fallopian tube cancers: A retrospective review.	36 ovarian cancer (+24 uterine cancer)	Retrospective review was performed inclusive of all patients who received ≥2 cycles of weekly ixabepilone (16–20 mg/m^2^ days 1, 8, 15 of a 28-day cycle) ± biweekly bevacizumab (10 mg/kg days 1 and 15).	Patients completed a mean of 4.7 ± 2.9 cycles of ixabepilone; 91.7% (33/36) of patients with ovarian cancers received concurrent bevacizumab. Weekly ixabepilone with or without biweekly bevacizumab has promising activity and acceptable toxicity in patients with platinum- or taxane-resistant endometrial and ovarian cancers.	Ixabepilone dose was reduced in patients with neuropathy and bevacizumab was reduced due to mucositis. Unacceptable toxicity in four patients included fatigue, proteinuria, neuropathy, diarrhea, mucositis, and new-onset seizures.
2021	Roque et al. [18]	Randomized phase II trial of weekly ixabepilone with or without biweekly bevacizumab for platinum-resistant or refractory ovarian, fallopian tube, primary peritoneal cancer.	78	Randomized to receive either ixabepilone monotherapy at 20 mg/m^2^ on days 1, 8, and 15 of a 28-day cycle, or ixabepilone at 20 mg/m^2^ plus bevacizumab at 10 mg/kg on days 1 and 15 of the same cycle.	PFS for ixabepilone plus bevacizumab was 5.5 months compared to 2.2 months for ixabepilone alone (*p* <0.001). OS was 10.0 and 6.0 months for the combination and monotherapy arms, respectively (*p* = 0.006). The ORR was 33% with ixabepilone plus bevacizumab vs. 8% with ixabepilone monotherapy (*p* = 0.004).	Both regimens were well tolerated.

* Dose dense. Abbreviations used: GOG (Gynecologic Oncology Group), BSA (Body-surface Area), AUC (Area Under the Curve), MTD (maximum-tolerated dose), EOC (epithelial ovarian cancer), ICON (International Collaborative Ovarian Neoplasm Group), IV (intravenous), SCOTROC (Scottish Randomised Trial in Ovarian Cancer), JGOG (Japanese Gynecologic Oncology Group), ORR (Objective response rate), RECIST (Response Evaluation Criteria in Solid Tumors), PLD (pegylated liposomal doxorubicin), SD (Stable Disease), PFS (progression free survival), OS (overall survival), AURELIA (Avastin Use in Platinum-Resistant Epithelial OC).

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
