# Peer review of "Microtubule-Interfering Drugs: Current and Future Roles in Epithelial Ovarian Cancer Treatment"

_cancers, 2021, doi:10.3390/cancers13246239_

Round 1

Reviewer 1 Report

In this revised manuscript, the authors have addressed the reviewer's suggestions and significantly improved the depth and details of the review article. The authors provide a comprehensive overview on how various microtubule inhibitors have been used in treatment of newly diagnosed and recurrent ovarian cancer. In the later section, the authors also provide valuable insights on common resistance mechanisms against microtubule inhibitors and future directions for the use of microtubule-interfering agents in ovarian cancer. In overall, this review article provides a very comprehensive and in-depth discussion on the use of microtubule-interfering agents in ovarian cancer, which will be very helpful for both clinicians and researchers in ovarian cancer community. Thus it is recommended for publication. Please fix a minor error in the title (Epi-Thelial to Epithelial) before publication. 

Reviewer 2 Report

Congratulation and thank you for your revision 

This manuscript is a resubmission of an earlier submission. The following is a list of the peer review reports and author responses from that submission.

Round 1

Reviewer 1 Report

It is appreciated for the opportunity to review this paper. In this article, the authors reviewed the role of microtubule-interfering cytotoxic agents used in treating ovarian cancer, including front-line treatment and recurrent disease. The authors also discussed the mechanism of resistance to microtubule-stabilizing agents, which is highly valuable. Overall, this is a well-written review article, which covers most of the scientific information on the topic, with precise English and scientifically proved direction. However, a few points need to be addressed by the authors:

  1. Line 75, the statement “etoposide is used in advanced, recurrent epithelial ovarian cancer,” requires further explanation. Etoposide is included in BEP (bleomycin, etoposide, cisplatin), a preferred regimen for malignant germ cell tumors, and rarely used in treating advanced, recurrent epithelial ovarian cancer. Only a few references can be found regarding oral etoposide for platinum-resistant and recurrent epithelial ovarian cancer. These references should be added if etoposide is mentioned. 
  2. In Table 1, the authors presented evidence for microtubule-interfering cytotoxic agents used to treat ovarian cancer. However, some studies were conducted in a first-line setting, and some were second-line treatments. These differences should be clearly stated. It is suggested to divide this table into two parts (role in front-line treatment and recurrence treatment).
  3. In Table 1, dose-dense chemotherapy should be highlighted. 
  4. In each table, the wording should be more concise to allow better readability. 
  5. Line 133, the authors stated that “there is no clear consensus on the role of dose-dense chemotherapy.” I totally agree with it. However, the authors can describe it in a more humble status, since some data from the real-world showed the dose-dense chemotherapy may be well tolerated and effective and can be considered as the primary systemic therapy. I recommended that the authors can take them into consideration. For examples: 

Huang CY, Cheng M, Lee NR, Huang HY, Lee WL, Chang WH, Wang PH. Comparing Paclitaxel-Carboplatin with Paclitaxel-Cisplatin as the Front-Line Chemotherapy for Patients with FIGO IIIC Serous-Type Tubo-Ovarian Cancer. Int J Environ Res Public Health. 2020 Mar 26;17(7):2213. doi: 10.3390/ijerph17072213. PMID: 32224896; PMCID: PMC7177627.

Cheng M, Lee HH, Chang WH, Lee NR, Huang HY, Chen YJ, Horng HC, Lee WL, Wang PH. Weekly Dose-Dense Paclitaxel and Triweekly Low-Dose Cisplatin: A Well-Tolerated and Effective Chemotherapeutic Regimen for First-Line Treatment of Advanced Ovarian, Fallopian Tube, and Primary Peritoneal Cancer. Int J Environ Res Public Health. 2019 Nov 29;16(23):4794. doi: 10.3390/ijerph16234794. PMID: 31795359; PMCID: PMC6926653.

6. Line 253-297, the authors described chemoresistance mechanisms of microtubule interfering cytotoxic agents, which is highly appreciated. However, an illustration or diagram of mechanisms in drug resistance is strongly recommended for this article. 

7. Additionally, is Figure 1 a direct copy from the referenced paper? If yes, it is suggested that the authors draw illustrations that summarize the current evidence.

Reviewer 2 Report

Overview

Tymon-Rosario et al.  have submitted a review of the use of microtubule-targeting drugs in epithelial ovarian cancer treatment. The review covers a large number of clinical trials, mostly using the standard platinum-based treatments (carboplatin or cisplatin) with a microtubule-targeting drug (paclitaxel or docetaxel) as a co-treatment. The manuscript concentrates mostly on recurrence of the cancer and development of resistance to the standard treatment drugs. The carboplatin/paclitaxel combination has become the standard treatment for epithelial ovarian cancer, but it has serious side effects, including neuropathy (paclitaxel) and neutropenia (docetaxel).  The authors conclude, because of acquisition of the resistance by the cancer and known side effects, that future therapies could involve a different microtubule-stabilising drug of the epothilone class which has advantages over the taxane drugs, namely not being substrates of drug efflux pumps, lack of susceptibility to taxane site mutations, less need for complex, hyperallergic diluents, and possibly fewer side effects. The manuscript is well-written, concise, and well-structured, making it easy for oncologists to obtain guidance for a good starting treatment for their patients or treatment after recurrence.

Major Comments

  1. The two tables summarise the clinical trials in detail, and the information is nicely packaged. The essential mechanisms of action of the taxane drugs are covered in the text, without getting into too much detail. Perhaps a patupilone (epothilone B) entry should be included in Table 1.a following the ixabepilone entry.
  2. In the Simple Summary and Abstract, the epothilone B synthetic variant ixabepilone is suggested as an alternative to paclitaxel or docetaxel. Then in Table 1.b, four separate clinical trials are summarized that used ixabepilone or patupalone. However, in the discussion, little is mentioned of the well-documented adverse side effects of ixabepilone in metastatic breast cancer, including neuropathy, neutropenia, myelosuppression, and hypersensitivity. How did ixabepilone compare with patupilone for adverse side effects?  Ixabepilone was FDA-approved in 2007 but not EMA approved, dropped by Bristol-Myers Squibb for safety reasons, and transferred to Allarity Therapeutics for further clinical development. Perhaps a brief addition to the section on “Role in front-line treatment” that covers some of the potential advantages of ixabepilone and or patupilone would help support the conclusion that ixabepilone (or patupilone?) should be considered as a replacement for paclitaxel in the platinum/taxane co-therapies.
  3. In the “Future Directions” section, a novel area is referred to, that of microtentacles seen in breast cancer cells and also found in ovarian cancer ascites fluid. The authors discuss the potential use of single-cell tethering to monitor the presence of microtentacles. It will be interesting to see how this research progresses.
  4. Despite including ixabepilone in the Simple Summary and Abstract, no mention is made of ixabepilone in the conclusion of the manuscript. This seems an unusual omission.
  5. In the ”Future Directions”, to broaden the scope and possibilities, a very brief statement of possible other microtubule-stabilising agent sites under investigation could be given, including the laulimalide/peloruside site. Different sites have different property profiles and could change the dynamics of treatment for ovarian cancer. Also, no mention is made of the depolymerising agents and whether it might be worth trialling them as well.

Minor Comments

  1. Line 87 Table 1a summarizes
  2. Line 88 and Table 1b illustrates
  3. In Table 1a, Teniposide entry, 2nd column        Podophyllum peltatum should be italicisized.
  4. In Table 1.a under ixabepilone, 4th column intracellular transport and inhibits mitotic spindle formation similar to the taxanes.
  5. In Table 1.b, McGuire entry ff, last column progression-free survival should be abbreviated as PFS. See also Piccart, and Sharma.
  6. In Table 1.b, Katsumata entry, last column     Don’t center the data and text for “the conventional…p=0.03).
  7. Line 130 with a platinum drug became the
  8. Line 212 drugs that have been shown to have a modest

Reviewer 3 Report

This review article provides an overview of the role of microtubule inhibiting drugs in the treatment of newly diagnosed and recurrent ovarian cancer. It is written clearly and is informative in general. However, this review is heavily focused on clinical results of paclitaxel, which is extensively reviewed elsewhere. Although there are numerous microtubule inhibitors listed in Table 1a, limited discussion is provided for these microtubule inhibitors in primary and recurrent ovarian cancer. For instance, is ixabepilone being considered for front line therapy? There are only 4 lines about abraxane (line 182-186). What are the efficacy and toxicity profile of abraxane compared to paclitaxel? Please expand this section. In addition, there are few biological and mechanistic information underlying the mechanism of action of microtubule inhibitors and the resistance mechanism. Thus, it would be helpful to provide more in-depth discussion about the biology before discussing clinical results. It would be better to start with basic biology covering the structure and functions of microtubes, the mechanism of action of various microtubule inhibitors (lines 255~263 can be moved to the first section), and the resistance mechanism. Regarding the resistance mechanisms, adding the discussion of the contribution of the tumor microenvironment to paclitaxel resistance would be informative. More studies also show that chemotherapeutic drugs can also cause cellular senescence and inflammation, thereby establishing the environment to cause more chemoresistance. It would be also informative to discuss how the second-generation microtubule inhibitors (e.g. ixabepilone) might overcome these resistance mechanisms.

After the authors discuss the role of microtubule inhibitors in primary and recurrent ovarian cancer, please add the descriptions of what clinical trials are currently ongoing with microtubule inhibitors.  

In Future Directions section, it would be helpful to discuss the current challenges of using microtubule inhibitors in clinics and how they can be addressed. For instance, how can the toxicity issues be addressed? What needs to be improved from the current microtubule inhibitor drugs? Are there any new studies that discover better biomarkers to predict therapy response or test new combination therapies with microtubule inhibitors? Are there any new microtubule inhibitors being developed?

Please cite the references more thoroughly. For example, please cite references at the end of each individual drug in lines 54~80. Please cite the references in line 101 (cisplatin with cyclophosphamide), line 129 (OV90 study), lines 198, 202 and 251 (the reference for 'tumor expression of class III beta-tubulin...did not predict response'). Also please add reference numbers to Table 1a and Table 1b.

In Table 1b, please include the name of clinical trials (e.g. GOG132, SCOTROC, GOG126M) and patient numbers enrolled in clinical trials. For the readers’ understanding, it might be better to split the key finding sections into two sections: clinical outcomes and toxicity/adverse events. It is not clear why Pfisterer et al (2018) is included in Table 1b since this trial did not use any microtubule inhibitors. At the end of Table 1b, it would be helpful to include a footnote to define abbreviations (e.g. ORR, PLD, SD, PFS, OS).

Line 121: ‘no superiority between the tree treatment arms’ In contrast to this statement, Table 1b says cisplatin alone or in combination yielded 'superior' response rates and PFS. Please correct it accordingly.

Line 212: Please provide some examples of single-agent cytotoxic drugs that show ORR 10-15% and their references.
